# INFANTNET: A LARGE SCALE DATASET FOR INFANT BODY POSE AND SHAPE ESTIMATION

## ABSTRACT

Infant pose and shape estimation is essential for applications in childcare, developmental monitoring, and medical diagnosis. However, existing methods and datasets are largely designed for adults, and direct transfer to infants fails due to substantial differences in body proportions, articulation limits, and frequent self-occlusion. To address this gap, we introduce InfantNet, the largest real-image infant dataset to date, comprising 108,902 RGB images of infants aged 6-18 months. Each image is annotated with 2D keypoints, and a curated subset of 11,642 images additionally includes 3D pose and shape annotations with full SMIL parameters. We use an iterative annotation pipeline to ensure high fidelity across both 2D and 3D labels. InfantNet establishes a large-scale, comprehensive benchmark for infant 2D keypoint detection and 3D pose-and-shape recovery. Baseline experiments demonstrate that state-of-the-art adult pose estimators do not generalize well to infants, whereas fine-tuning on InfantNet yields a consistent improvement. The gains are even more pronounced for 3D pose and shape estimation. By releasing the InfantNet dataset and benchmark, we provide a vital resource for advancing infant pose analysis and related healthcare applications.

## 1 INTRODUCTION

Estimating the pose and shape of the human body is a foundational task in computer vision, with broad applications in action recognition, human-computer interaction, and healthcare (Kanazawa et al., 2018; Kolotouros et al., 2019; Pavlakos et al., 2019; Nachman et al., 2024; Lupolt et al., 2025). Over the past decades, human pose and shape estimation algorithms have achieved remarkable progress. However, existing methods and datasets predominantly focus on adult subjects, whose body morphology and movement patterns differ significantly from those of infants. Directly applying adult-trained models to infants results in poor performance due to substantial domain gaps, including differences in body proportions, limited articulation, and frequent self-occlusion typical of infant behavior. These limitations significantly hinder the deployment of such models in infant-related applications, despite their critical importance in areas such as developmental monitoring, childcare, and early medical diagnosis.

A major obstacle to advancing infant-specific pose estimation algorithms is the lack of large-scale, high-quality annotated datasets tailored to infants. Infant datasets typically contain only a few thousand images (Hesse et al., 2018a; Huang et al., 2021; Dechemi et al., 2021), in contrast to adult datasets such as 3DPW(von Marcard et al., 2018) and Human3.6M(Ionescu et al., 2014), which comprise over 51,000 and 3.6 million images, respectively. Existing datasets are either limited in scale, synthetic in nature, or focused primarily on 2D pose estimation with sparse annotations (Hesse et al., 2018a; Huang et al., 2021; Dechemi et al., 2021), making them inadequate for training or evaluating state-of-the-art 2D and 3D human modeling methods (Xu et al., 2022; Goel et al., 2023).

To address this gap, we introduce **InfantNet**, a large-scale dataset specifically designed for infant pose and shape estimation. Comprising more than 100K high-resolution RGB images of infants mainly aged 6 to 18 months, InfantNet includes accurate annotations of 17 anatomically consistent 2D keypoints. A carefully curated subset of 11,642 images further includes 3D pose and shape annotations by SMIL (Hesse et al., 2018b) parameters, enabling future research for 3D infant shape and pose estimation. We adopt a robust iterative annotation pipeline that combines human supervision with optimization-based refinement to ensure high-quality labels in 2D. We further validate 3D

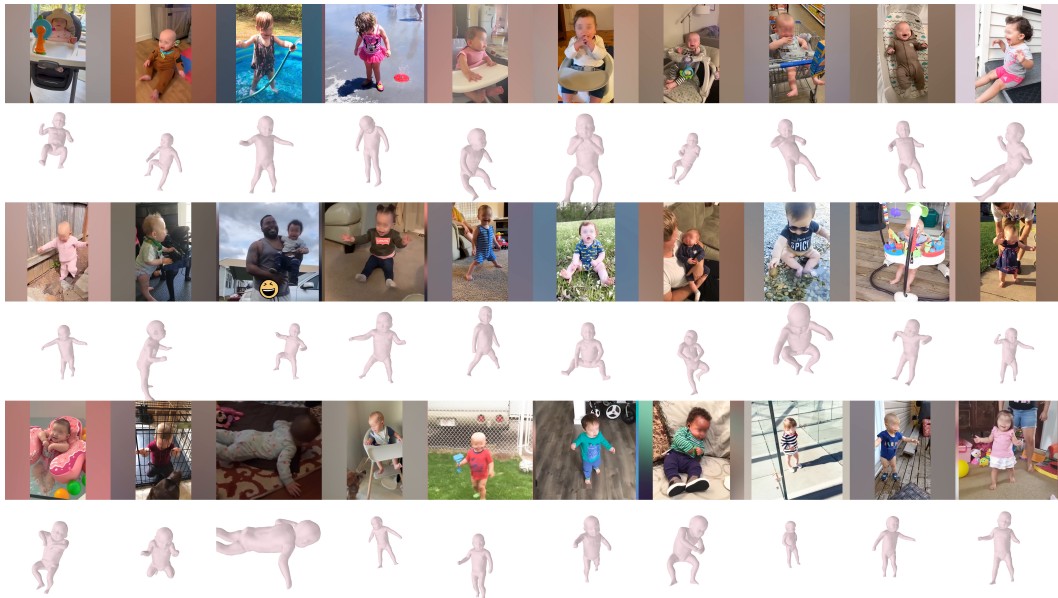

Figure 1: Sample SMIL fitting results from the proposed InfantNet dataset. Our dataset contains a wide range of shapes and poses of infants with high-quality 2D and 3D annotations.

SMIL (Hesse et al., 2018b) pose-and-shape parameters by predicting the resulting meshes from multiple viewpoints. Examples of images and 3D pose and shape annotations are shown in Figure 1.

Our comprehensive benchmark shows that existing state-of-the-art adult pose estimators generalize poorly to infants. In contrast, fine-tuning on InfantNet yields a substantial boost in performance across key metrics such as Average Precision. We further adapt parametric 3D human models and learning frameworks to the infant domain, narrowing the gap between infant and adult modeling accuracy. By training 3D human pose and shape estimation models on our infant dataset, we are able to achieve performance comparable to training on much larger adult datasets with precise 3D ground truth.

By releasing InfantNet along with standardized evaluation protocols and baseline models, we aim to catalyze future research in infant-centric vision tasks. We believe our work constitutes a critical step toward enabling scalable, robust, and accurate infant pose and shape analysis in naturalistic environments, with implications for early developmental assessment, clinical monitoring, and beyond. In summary, the main contributions of our paper are:

- We introduce InfantNet, a large-scale dataset specifically designed for infant pose and shape estimation. It comprises 108,902 RGB images of infants with high-quality 2D keypoint annotations. A curated subset of 11,642 images further includes 3D ground truth pose and shape parameters, enabling comprehensive evaluation of both 2D and 3D methods.
- We develop an end-to-end SMIL-based pipeline for infant 3D pose and shape estimation. To our knowledge, this is the first attempt to directly regress SMIL parameters from images with neural networks. We generate pseudo ground-truth labels via optimization-based SMIL fitting, followed by manual verification and filtering to ensure high-quality supervision. A modified human 3D pose and shape regression network is then trained on our InfantNet, yielding end-to-end models specialized for infant SMIL parameter estimation.
- We analyze the generalization gap of adult pose and shape estimators on infant data. Fine-tuning 2D keypoint detectors on InfantNet yields clear performance gains, highlighting the need for domain adaptation. Training end-to-end SMIL-based 3D pose and shape models on InfantNet further achieves performance comparable to adult benchmarks.
- We set up a benchmark on InfantNet using state-of-the-art human pose estimation algorithms. By releasing data, code, and evaluation protocols, we hope to provide a foundation and standard for future research in infant pose and shape estimation.

| Dataset | Syn/Real | # Images | Keypoints | Segmentation | 3D Annotations |
|---|---|---|---|---|---|
| MINI-RGBD (Hesse et al., 2018a) | Synthetic | 12,000 | Yes | No | Only depth |
| SyRIP (Huang et al., 2021) | Syn+Real | 1,700 | Yes | No | No |
| InfantNet (ours) | Real | 108,902 | Yes | Yes | Yes |

Table 1: Comparison of InfantNet with existing open-source infant keypoint datasets. InfantNet contains over 100,000 real-world infant images with high-quality keypoint annotations, significantly surpassing existing datasets in both scale and quality. In addition, we provide segmentation masks and bounding boxes for all images, along with a curated subset annotated with 3D SMIL parameters.

## 2 RELATED WORK

In this section, we discuss existing infant datasets and related pose and shape estimation algorithms.

### 2.1 INFANT DATASETS

Human pose and shape estimation has benefited enormously from large-scale adult datasets such as Human3.6M (Ionescu et al., 2014), MPII (Andriluka et al., 2014), and AMASS (Mahmood et al., 2019). In contrast, infant datasets remain scarce due to ethical, logistical, and technical challenges in capturing infant motion in naturalistic environments. Early efforts include BabyPose (Migliorelli et al., 2020), which provides 16 depth videos from NICU settings with 12-joint annotations, and the dataset from (Groos et al., 2022), focusing on supine infants with limited poses. BabyNet (Dechemi et al., 2021) targets activity classification without fine-grained pose labels. MINI-RGBD (Hesse et al., 2018a) offers synthetic depth sequences of 12 infant models, while SyRIP (Huang et al., 2021) combines 1,700 synthetic and real images with sparse 2D keypoints. These datasets are small, often synthetic or modality-specific, and lack segmentation masks or 3D ground truth. Our proposed *InfantNet* addresses these limitations by providing real infant RGB images with high-quality 2D keypoints, segmentation masks, bounding boxes, and 3D pose and shape annotations. Table 1 compares InfantNet with prior datasets, showing its unique scale, diversity, and multimodal supervision, which together enable robust evaluation and training of infant-specific models.

### 2.2 INFANT POSE AND SHAPE ESTIMATION

2D infant pose estimation initially applied general-purpose models such as OpenPose (Cao et al., 2019), HRNet (Sun et al., 2019), and AlphaPose (Fang et al., 2022), but these perform poorly due to infants' distinct proportions, articulation limits, and frequent self-occlusion (Sciortino et al., 2017). Later work fine-tuned these CNN-based estimators on small infant datasets (Groos et al., 2022), reducing error and approaching human inter-rater variability. More recent transformer-based methods, such as ViTPose (Xu et al., 2022) and AggPose (Cao et al., 2022), improve spatial reasoning in cluttered settings (Jahn et al., 2025), but require larger, more diverse datasets to avoid overfitting.

In the 3D domain, early infant methods relied on depth sensors for direct joint prediction (Hesse et al., 2018a) or stereo systems for triangulation (Soualmi et al., 2024), achieving good accuracy in clinical settings but requiring specialized hardware. The SMIL model (Hesse et al., 2018b) represents a key step toward parametric modeling of infants: adapting the adult SMPL template (Loper et al., 2015) with an infant-specific mesh and priors learned from noisy, incomplete RGB-D recordings. While SMIL enables full-body mesh registration, its reliance on limited and low-quality depth sequences constrains generalization. Other approaches leverage synthetic renderings (Huang et al., 2021) or alternative sensors such as pressure mats (Donati et al., 2013; Clever et al., 2020), but these are less practical in everyday environments.

Despite this progress, the field lacks a large-scale, real-world RGB benchmark with consistent 2D and 3D annotations. Our *InfantNet* fills this gap, enabling systematic development and evaluation of infant pose and shape estimation models that generalize across naturalistic scenarios.

### 2.3 ADULT POSE AND SHAPE ESTIMATION

3D human pose and shape estimation has been extensively studied through parametric models such as SMPL (Loper et al., 2015) and its extensions (e.g. SMPL-X (Pavlakos et al., 2019)). Learning-based

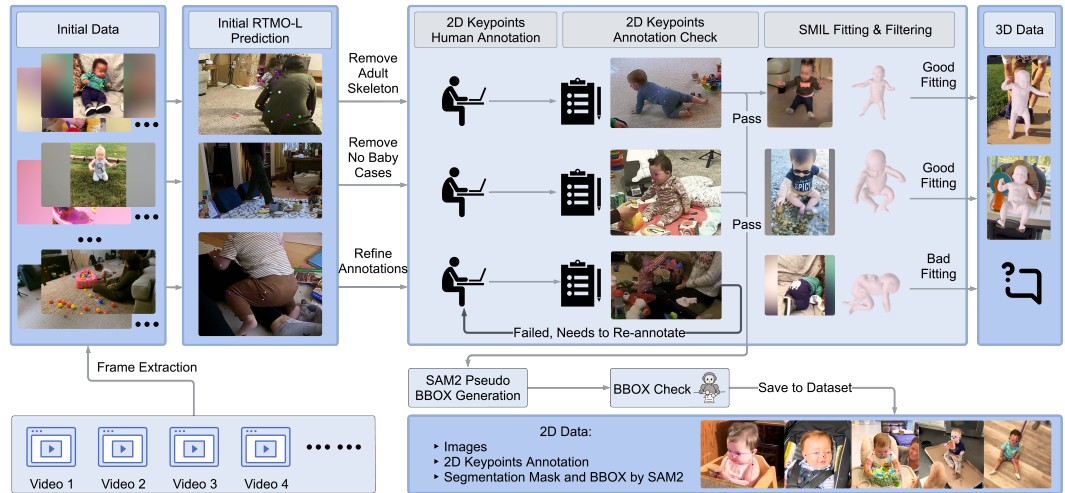

Figure 2: Annotation pipeline of our InfantNet Dataset. Frames from YouTube and HomeVisit recordings were annotated using a semi-automatic, inspector-guided workflow. For 2D labels, RTMO-L provided initial keypoints that were refined by annotators with iterative quality control, following the COCO format. For 3D labels, we applied an optimization-based method to fit SMIL to infant images, and annotators further filtered the results to ensure accurate pose and shape annotations.

approaches including HMR (Kanazawa et al., 2018), VIBE (Kocabas et al., 2020), PARE (Kocabas et al., 2021), and 4D-Humans (Goel et al., 2023) directly regress parameters from images or videos, often with strong temporal consistency. However, these methods are trained on adult datasets and exhibit poor generalization to infants, whose body proportions, articulation limits, and frequent occlusions differ significantly. In this work, we adapt such adult regression networks to the infant domain using InfantNet and SMIL-based annotations.

## 3 INFANTNET DATASET

In this section, we present the InfantNet dataset and detail the data collection and annotation procedures. All data collection was conducted under Institutional Review Board (IRB) approval, with potential risks clearly disclosed to participants during home visits.

### 3.1 DATA COLLECTION

**YouTube data collection.** We curated publicly available infant videos from YouTube, capturing infants in a wide range of poses and body shapes. These raw videos serve as one source of our dataset and are later processed through annotation and quality filtering.

**HomeVisit data collection.** We collected multi-view video recordings of infants in daily home environments. Each session was conducted in the participant's primary play area, where we deployed four statically mounted GoPro Hero9 cameras positioned orthogonally at the walls, overlapping coverage of the scene. To enable accurate SMIL fitting, we conducted rigorous per-session camera calibration. Intrinsic calibration was performed independently for each camera using a small planar checkerboard (6×10 squares), tilting and sweeping it to capture lens distortion and intrinsic parameters. Extrinsic calibration was then performed using a large checkerboard (12×8 squares) simultaneously visible to all four cameras, allowing estimation of relative poses via multi-view bundle adjustment. Calibration was repeated both before and after each session to ensure pose stability. Videos were recorded at 1920×1080 resolution and 60 FPS, yielding over 100,000 time-synchronized RGB frames. After data collection, we post-process videos from four cameras for video synchronization. Written consent was obtained from all caregivers.

Original Image   Segmentation   BBox And Keypoints   Original Image   Segmentation   BBox And Keypoints

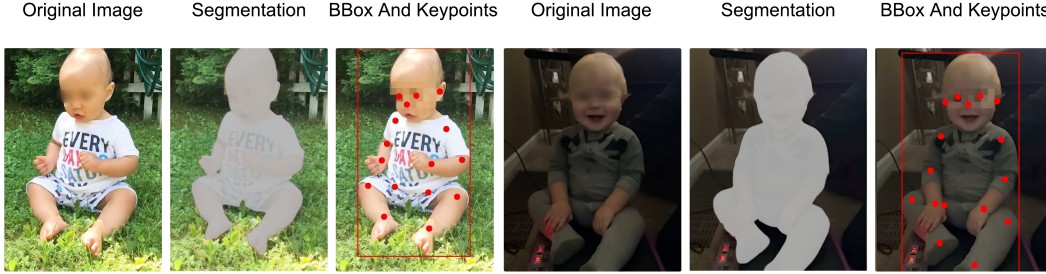

Figure 3: Visualization of different annotation stages in our InfantNet Dataset: Original image, Segmentation mask, Bounding box, and Annotated Body Keypoints.

## 3.2 DATA ANNOTATION

We annotated all frames extracted from the two sources described above, while ensuring that each retained frame contained exactly one infant. In total, 206,436 frames were extracted prior to quality filtering. For the YouTube subset, which consists of 10 videos, we obtained 51,039 frames. Because these videos are often edited compilations featuring multiple different infants, we discarded frames containing more than one infant, retaining only single-infant frames for annotation. For the HomeVisit subset, which includes 15 multi-camera sessions (60 synchronized video streams), we extracted 155,397 frames. Each session involves only one infant, so no additional filtering was required, though temporal alignment across the four synchronized cameras was enforced. After frame selection, we carried out 2D body keypoint annotation followed by multi-stage quality control, removing frames with severe occlusion, motion blur, or other degradations. The resulting high-quality annotations serve as the ground truth supervision for SMIL fitting and subsequent model training.

To annotate such a large and heterogeneous dataset, we adopted a semi-automatic, inspector-guided annotation pipeline, inspired by human-in-the-loop strategies in 3D model fitting works such as Animal3D (Xu et al., 2023). As shown in Figure 2, our annotation workflow consisted of six stages designed to ensure both efficiency and accuracy.

**Initial keypoint estimation.** We first employed the RTMO-L model (Lu et al., 2024) from MM-Pose (Contributors, 2020) to produce pseudo ground truth 2D keypoints for all frames. These predictions were loaded into CVAT Contributors (2018), where human annotators conducted multi-step refinement.

**Initial filtering and manual correction.** Annotators reviewed the RTMO-L (Lu et al., 2024) predictions in CVAT and removed frames with severe motion blur, camera shake, strong occlusion (e.g., when the infant is mostly outside the field of view or heavily covered by objects), missing infants, or subjects older than 16 months. Frames with strong lighting artifacts, such as overexposure or harsh shadows, were also excluded. Adult skeletons were discarded, and only one infant per frame was retained. For the remaining frames, if the RTMO-L prediction provided accurate and anatomically plausible keypoints, the annotation was accepted without modification to maximize efficiency. Otherwise, annotators manually corrected the predictions to ensure anatomical consistency. All annotations followed the COCO format with 17 joints: nose, left eye, right eye, left ear, right ear, left shoulder, right shoulder, left elbow, right elbow, left wrist, right wrist, left hip, right hip, left knee, right knee, left ankle, and right ankle. Each keypoint was tagged as *visible*, *occluded*, or *not present*.

**Inspector verification.** An inspector reviewed the annotations to ensure consistency and quality. Frames with annotation errors were returned to annotators with revision feedback.

**Iterative refinement.** This correction–verification cycle was repeated until the inspector approved all annotations for each recording session, ensuring convergence to high-quality labels.

**Bounding box generation.** Once the keypoints were finalized, bounding boxes were generated using SAM2 (Ravi et al., 2024). Visible keypoints were used to prompt mask generation, and the final bounding boxes were computed as the minimal rectangles covering both the masks and any occluded joints, thereby mitigating segmentation errors due to partial occlusion (e.g., infants partially covered by furniture).

| Subset | Viewpoint Info | Raw Frames | Filtered Frames |
|---|---|---|---|
| YouTube (Train) | Varied | 44,257 | 36,185 |
| YouTube (Test) | Varied | 6,782 | 3,959 |
| HomeVisit (Train) | 4 fixed cameras | 123,876 | 56,968 |
| HomeVisit (Test) | 4 fixed cameras | 31,521 | 11,790 |
| **Total** | – | 206,436 | 108,902 |

Table 2: Summary of dataset composition and frame filtering. The frame count after filtering reflects the removal of low-quality, irrelevant, or ambiguous samples during annotation.

**Final bounding box validation.** In the final stage, inspectors performed a second review to verify the accuracy and tightness of bounding boxes, ensuring annotation completeness and quality.

This staged workflow allowed us to balance scalability and precision, producing high-quality 2D annotations with explicit human oversight at each critical step. Examples of the 2D annotations in different stages are shown in Figure 3.

### 3.3 FITTING SMIL ON INFANT IMAGES

We use an optimization-based method to fit SMIL (Hesse et al., 2018b) on infant images from our dataset following the FiDIP (Huang et al., 2021) workflow. We optimize the shape parameters $\beta$ and pose parameters $\theta$ in SMIL by minimizing the total loss:

$$\mathcal{L}_{\text{SMIL}} = \mathcal{L}_{\text{data}} + \mathcal{L}_{\text{pose}} + \mathcal{L}_{\text{shape}} + \mathcal{L}_{\text{others}}. \tag{1}$$

The data term, $\mathcal{L}_{\text{data}}$, includes mesh-to-scan $\mathcal{L}_{\text{m2s}}$ and scan-to-mesh $\mathcal{L}_{\text{s2m}}$ distances. $\mathcal{L}_{\text{pose}}$ and $\mathcal{L}_{\text{shape}}$ enforce priors on $\theta$ and $\beta$, respectively. $\mathcal{L}_{\text{others}}$ contains losses for landmark keypoints alignment ($\mathcal{L}_{\text{lm}}$), table contact ($\mathcal{L}_{\text{table}}$), temporal smoothness ($\mathcal{L}_{\text{sm}}$), and self-intersections ($\mathcal{L}_{\text{sc}}$).

The registration optimization proceeds in three steps: estimating initial shape from selected frames, solving per-frame poses with smoothness and data terms, and registering surface meshes via constrained deformation. The resulting meshes are validated by multi-view visualization and a 16px reprojection error threshold. The filtered pose and shape parameters provide high-quality 3D ground-truth annotations for training end-to-end models.

### 3.4 DATA SUMMARY

We partition the dataset into training and testing splits while maintaining identity consistency across videos and sessions. Table 2 reports the statistics of raw frame counts and the final numbers after annotation-stage filtering.

**YouTube Subset.** The YouTube subset consists of 10 public videos with diverse environments and camera conditions. Two videos (**3,959 frames**) were designated for testing, while the remaining **36,185 frames** were assigned to training. The difference between raw and filtered counts reflects the removal of multi-infant frames and other ineligible samples during annotation.

**HomeVisit Subset.** The HomeVisit subset contains 15 in-home recording sessions, producing a total of 155,397 frames. For evaluation, we selected 3 sessions as the testing set, while the remaining 12 sessions were assigned to training. After annotation-stage filtering, the testing set contains **11,790 frames** and the training set contains **56,968 frames**.

**Format and Release.** All annotations follow the standard **COCO keypoints format** (Lin et al., 2014) and include 17 body keypoints with visibility flags (visible, occluded, not present), as well as bounding boxes. The dataset will be released publicly to support future research in infant-specific human pose estimation. Its multi-source, multi-view, and age-diverse composition enables both robust training and principled benchmarking in this challenging domain.

| Model | Direct Inference | | | | Fine-tuned | | | |
|---|---|---|---|---|---|---|---|---|
| | AP | AP.5 | AP.75 | AR | AP | AP.5 | AP.75 | AR |
| Hourglass 256x256 | 0.8627 | 0.9558 | 0.9237 | 0.8843 | 0.9263 | 0.9899 | 0.9790 | 0.9395 |
| Hourglass 384x384 | 0.8739 | 0.9659 | 0.9343 | 0.8927 | 0.9340 | 0.9898 | 0.9789 | 0.9452 |
| HRFormer-small_256x192 | 0.8765 | 0.9653 | 0.9336 | 0.8982 | 0.9314 | 0.9897 | 0.9787 | 0.9436 |
| HRFormer-small_384x288 | 0.8800 | 0.9654 | 0.9430 | 0.9007 | 0.9351 | 0.9897 | 0.9791 | 0.9460 |
| HRFormer-base_256x192 | 0.8824 | 0.9550 | 0.9333 | 0.9026 | 0.9385 | 0.9898 | 0.9792 | 0.9509 |
| HRFormer-base_384x288 | 0.8871 | 0.9638 | 0.9416 | 0.9082 | 0.9401 | 0.9898 | 0.9793 | 0.9527 |
| Res50-256x192 | 0.8648 | 0.9662 | 0.9337 | 0.8859 | 0.9273 | 0.9900 | 0.9791 | 0.9404 |
| Res50-384x288 | 0.8657 | 0.9658 | 0.9331 | 0.8863 | 0.9303 | 0.9900 | 0.9793 | 0.9416 |
| Res101-256x192 | 0.8728 | 0.9656 | 0.9344 | 0.8929 | 0.9318 | 0.9899 | 0.9790 | 0.9437 |
| Res101-384x288 | 0.8759 | 0.9650 | 0.9338 | 0.8957 | 0.9351 | 0.9899 | 0.9792 | 0.9478 |
| Res152-256x192 | 0.8731 | 0.9556 | 0.9341 | 0.8938 | 0.9331 | 0.9898 | 0.9792 | 0.9457 |
| Res152-384x288 | 0.8789 | 0.9646 | 0.9333 | 0.8999 | 0.9387 | 0.9895 | 0.9789 | 0.9490 |
| HRNet-w32_dark_256x192 | 0.8860 | 0.9661 | 0.9348 | 0.9037 | 0.9382 | 0.9898 | 0.9792 | 0.9489 |
| HRNet-w32_dark_384x288 | 0.8874 | 0.9650 | 0.9429 | 0.9064 | 0.9398 | 0.9897 | 0.9792 | 0.9519 |
| HRNet-w48_dark_256x192 | 0.8892 | 0.9656 | 0.9440 | 0.9069 | 0.9416 | 0.9896 | 0.9791 | 0.9528 |
| HRNet-w48_dark_384x288 | 0.8874 | 0.9651 | 0.9432 | 0.9060 | 0.9421 | 0.9898 | 0.9791 | 0.9534 |
| HRNet-w32_256x192 | 0.8804 | 0.9660 | 0.9345 | 0.8995 | 0.9336 | 0.9898 | 0.9788 | 0.9450 |
| HRNet-w32_384x288 | 0.8819 | 0.9653 | 0.9339 | 0.9024 | 0.9364 | 0.9896 | 0.9788 | 0.9482 |
| HRNet-w48_256x192 | 0.8838 | 0.9657 | 0.9341 | 0.9024 | 0.9374 | 0.9898 | 0.9792 | 0.9485 |
| HRNet-w48_384x288 | 0.8837 | 0.9650 | 0.9337 | 0.9039 | 0.9397 | 0.9897 | 0.9790 | 0.9512 |
| ViTPose-small_256x192 | 0.8756 | 0.9656 | 0.9347 | 0.8946 | 0.9336 | 0.9899 | 0.9793 | 0.9443 |
| ViTPose-base_256x192 | 0.8902 | 0.9654 | 0.9448 | 0.9089 | 0.9391 | 0.9897 | 0.9791 | 0.9511 |
| ViTPose-large_256x192 | **0.9002** | 0.9644 | 0.9434 | **0.9199** | 0.9475 | 0.9896 | 0.9791 | 0.9581 |
| ViTPose-huge_256x192 | 0.8995 | 0.9639 | 0.9430 | **0.9199** | **0.9477** | 0.9895 | 0.9790 | **0.9582** |

Table 3: Comparison of 2D pose estimation models on the InfantNet benchmark under two evaluation settings: direct inference using COCO-pretrained models (left block) and fine-tuning on InfantNet (right block). We report Average Precision (AP), AP at IoU thresholds 0.5 (AP.5) and 0.75 (AP.75), and Average Recall (AR). Fine-tuning consistently improves performance across all architectures, with ViTPose-huge achieving the best results after adaptation (AP: **0.9477**, AR: **0.9582**). These results highlight the importance of domain-specific data for accurate infant pose estimation.

## 4 EXPERIMENTS

### 4.1 2D KEYPOINT ESTIMATION

We benchmark a series of widely used 2D human pose estimation models on our proposed infant dataset to evaluate their ability to transfer across domains. These models were originally designed and trained for detecting keypoints in adults; here, we investigate whether they can generalize to detecting keypoints for infants, whose body morphology and movement patterns differ substantially. This evaluation provides the first set of baselines for infant 2D keypoint estimation in the wild.

**Setup.** We benchmark 24 models across three architecture families: Transformer-based (ViT-Pose (Xu et al., 2022), HRFormer (Yuan et al., 2021)), CNN-based (HRNet (Sun et al., 2019), Hourglass (Newell et al., 2016)), and CNN variants with DARK post-processing (HRNet-Dark (Zhang et al., 2020)). All models are implemented in MMPose and initialized from COCO-pretrained weights. We fine-tune each model for 40 epochs using the MMPose default setting with a learning rate of $1 \times 10^{-4}$. Training and evaluation are conducted on 4 NVIDIA A100-SXM4-40GB GPUs.

**Evaluation.** Models are fine-tuned on the mixed training set (YouTube$_{\text{Train}}$ ∪ HomeVisit$_{\text{Train}}$) and evaluated on the mixed test set (YouTube$_{\text{Test}}$ ∪ HomeVisit$_{\text{Test}}$). We do not run source-specific evaluations; all numbers are reported on the mixed test split. The split preserves infant identity separation across sessions. Metrics follow the standard COCO keypoint protocol over 17 keypoints: mean Average Precision (AP), AP.5, AP.75, and Average Recall (AR). We report results from the checkpoint that achieve the best validation AP.

**Results.** We benchmark a wide range of 2D human pose estimation models on InfantNet, comparing their performance under two settings: (1) direct inference using COCO-pretrained weights without adaptation, and (2) fine-tuning the models on our InfantNet training set. Table 3 summarizes the Average Precision (AP), Average Recall (AR), and their variants across 24 different configurations.

*Direct Inference Results.* ViTPose-large achieves the highest AP (0.9002), while ViTPose-large and ViTPose-huge reach the best AR (∼0.9199). This highlights the advantage of transformer-based models over CNN-based ones on infant data, though their performance remains below that of fine-tuned counterparts. Among CNN-based models, HRNet-W48 (Sun et al., 2019) and HRFormer-Base (Yuan et al., 2021) perform competitively, both benefiting from multi-resolution feature aggregation. The performance gap between small and large variants is consistent across architectures, with larger models showing greater gains from fine-tuning.

*Fine-tuning on InfantNet.* Fine-tuning consistently boosts performance across all models. On average, fine-tuning yields a relative gain of **6.3% in AP** and **5.3% in AR** compared to direct inference. The ViTPose-huge model again achieves the highest overall performance (AP: 0.9477, AR: 0.9582) after fine-tuning. Overall, models with higher input resolution outperform their lower-resolution counterparts, demonstrating the importance of spatial detail for infant pose estimation.

## 4.2 3D Pose and Shape Estimation

We further benchmark representative 3D pose and shape estimation models on our proposed infant dataset to assess their generalizability to infant body models, establishing baselines for 3D infant pose and shape estimation.

**Setup.** While there are no baselines explicitly designed for 3D infant pose and shape estimation, human pose and shape estimation models can be adapted to infants. We benchmark HMR (Kanazawa et al., 2018) and PARE (Kocabas et al., 2021) from CNN-based architectures, and 4D-Humans (Goel et al., 2023) based on transformers as baselines for human pose and shape estimation. We train each model for 100 epochs using $2.5 \times 10^{-5}$ as learning rate for HMR (Kanazawa et al., 2018) and PARE (Kocabas et al., 2021) on 3 NVIDIA TITAN RTX GPUs, and a learning rate of $2.0 \times 10^{-5}$ for 4D-Humans on 4 NVIDIA A5000 GPUs.

**Evaluation.** We evaluate models with standard 3D human pose and shape estimation metrics. We report mean per joint position error (MPJPE) in mm, which measures the absolute error between predicted and ground-truth 3D joint positions and Procrustes-aligned mean per joint position error (PA-MPJPE) in mm, which evaluates relative joint configuration error after rigid alignment.

**Model & Data Preparation.** We train the models with **10,071 frames** and evaluate with **1,571 frames**. We guide HMR (Kanazawa et al., 2018) and 4D-Humans (Goel et al., 2023) training with 2D and 3D supervisions. For PARE (Kocabas et al., 2021) training, we manually segment all body mesh vertices into 7 body parts for part-based attention and segmentation mask supervision.

**Results.** We benchmark 3D human pose and shape estimation models on InfantNet, assessing their performance under two experimental settings: (1) train and inference using model weights adapted to SMIL infant body model, and (2) direct inference using pretrained model weights for adult body model on our InfantNet validation set. Figure 4 presents a visualized comparison of end-to-end models trained on infant-specific data versus directly inferred using the adult body model on our dataset. Table 4 summarizes the mean per joint position error (MPJPE) and Procrustes-aligned mean per joint position error (PA-MPJPE) across 6 different model configurations.

*Adult Body Model Results.* Direct inference using human pose and shape estimation models yields lower accuracy compared to results obtained with infant-specific body models, affirming the domain gap between adult and infant poses and shapes. Among adult-centric models, PARE (Kocabas et al., 2021) achieves the strongest performance (MPJPE: 364.49 mm, PA-MPJPE: 120.06 mm). Nevertheless, its accuracy remains considerably lower than when trained with infant body models.

*Infant Body Model Results.* Adapting the human pose and shape estimation models to the infant-specific body model SMIL (Hesse et al., 2018b) leads to substantial performance improvement across all architectures. On average, there is a **78% reduction in MPJPE and 44% reduction in**

Figure 4: Visualization of 3D pose and shape estimation model performances. The columns from left to the input image, direct inference with adult body model and data, and training with infant body model and data for HMR (Kanazawa et al., 2018), PARE (Kocabas et al., 2021), and 4D-Humans (Goel et al., 2023), respectively.

| Model | Adult Body Model | | Infant Body Model | |
|---|---|---|---|---|
| | MPJPE↓ | PA-MPJPE↓ | MPJPE↓ | PA-MPJPE↓ |
| HMR (Kanazawa et al., 2018) | 393.30 | 124.15 | 103.36 (↓74%) | 58.79 (↓53%) |
| PARE (Kocabas et al., 2021) | 364.49 | 120.06 | **83.73** (↓77%) | 63.16 (↓47%) |
| 4D-Humans (Goel et al., 2023) | 526.62 | 125.02 | 84.73 (↓84%) | **46.74** (↓63%) |

Table 4: Comparison of 3D pose and shape estimation models on the InfantNet benchmark under two settings: direct inference with adult body models (left) vs. training and inference with infant body models (right). Metrics: MPJPE and PA-MPJPE (mm). Using infant body models yields consistent gains across all architectures, underscoring the importance of infant-specific data and models for accurate 3D estimation.

**PA-MPJPE** compared to direct inferencing with the adult body model. Our results for estimating 3D pose and shape parameters using the SMIL (Hesse et al., 2018b) body model for infants are almost as good as those obtained for adult body model and datasets (e.g., HMR (Kanazawa et al., 2018) on Human3.6M dataset reports MPJPE of 87.97 mm). Among the models evaluated, 4D-Humans (Goel et al., 2023) achieves the strongest results (MPJPE: 84.73 mm, PA-MPJPE: 46.74 mm), indicating that the structure of the vision transformers is more resistant to domain changes.

# 5 CONCLUSION

We introduce InfantNet, the first large-scale dataset specifically designed for infant pose and shape estimation, addressing the shortcomings of adult-centric models in this domain. With high-quality 2D keypoint annotations and a curated subset of 3D SMIL parameters, InfantNet enables both keypoint detection and full-body mesh recovery. Our benchmark shows that models trained on adult data generalize poorly to infants, while training and fine-tuning on InfantNet leads to substantial performance improvements. While a limitation is that acquiring 3D annotations is more resource-intensive than 2D labeling, our iterative fitting pipeline provides a reliable set of high-quality 3D labels. We are actively expanding the dataset and refining the 3D fitting process to further improve coverage and accuracy. InfantNet offers promising potential for applications in early developmental monitoring and rehabilitation, supporting progress in pediatric healthcare and research. To promote responsible use, considering privacy concerns and potential misuse of sensitive infant data, we mask facial regions and ensure all data was collected under appropriate consent and oversight. By releasing the dataset, annotations, and standardized evaluation tools, we hope to accelerate innovation in infant-focused vision models and foster meaningful impact in both academic and clinical settings.

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

# A APPENDIX

## A.1 USE OF LANGUAGE MODELS

Large language models (LLMs) were employed to assist with grammar refinement, phrasing improvements, and consolidation of text for clarity and readability. All technical content, experiments, results, and conclusions were developed and validated by the authors, and the scientific substance of the work was not influenced by LLM usage.

