# OpenReview forum: "InfantNet: A Large Scale Dataset for Infant Body Pose and Shape Estimation"
_ICLR.cc/2026/Conference — ICLR 2026 Conference Withdrawn Submission_

### Official Review · Reviewer_tEE1 · 2025-10-28

**Soundness:** 3
**Presentation:** 3
**Contribution:** 2
**Rating:** 4
**Confidence:** 4

**Summary:**

This paper introduces InfantNet, a dataset of 108,902 RGB images of infants aged 6-18 months with 2D keypoint annotations, and 11,642 images with 3D SMIL parameters. The labeling pipeline consists of predicting 2D Body Joints and 3DMM fitting with a human-in-the-loop. The authors benchmark state-of-the-art pose estimation models, demonstrating performance gaps when applied to infants versus adults.

**Strengths:**

- Addresses an important gap: there are a few images of infants in existing human body modeling datasets, and this work attempts to fill that void. Healthcare is a good example of a field that benefits from this data.
- Scale: The dataset is substantially larger than existing infant datasets.
- Multi-modal annotations: Provides 2D keypoints, segmentation masks, bounding boxes, and 3D SMIL parameters.
- Comprehensive benchmarking: Tests 24 2D pose estimation models and 3 3D models with detailed evaluation.

**Weaknesses:**

1. **Ethical Issues**
The most significant concern is the data collection methodology. While HomeVisit data was collected with IRB approval and informed consent, the YouTube subset raises serious ethical concerns such as : Lack of consent: Scraping infant images from YouTube videos without explicit consent for academic research use is ethically problematic, even if videos are publicly available; Privacy violations: Infants cannot consent to their data being used for ML research. The paper mentions face masking, but this is insufficient to remove all privacy-sensitive information. Precedent: There's a reason large infant datasets don't exist - due to ethical and privacy concerns. This work may set a concerning precedent for future research.
2. **Experimental Design (Table 3).** The evaluation setup is problematic. Training on (YouTube + HomeVisit) and testing on (YouTube + HomeVisit) doesn't properly evaluate generalization. Current setup inflates performance: Models may be memorizing scene characteristics or specific infants rather than learning robust pose estimation. It would be better to at least split the sources, for example Train on YouTube only and Test on HomeVisit (or vice versa) to assess cross-domain generalization.
3. **Experimental Design (Table 4).** The 3D comparison with adult SMPL models is not a fair baseline and is not insightful - it is expected that SMPL (adult model) fails on infants. A more appropriate baseline would be:
- Run SOTA 2D keypoint detector
- Fit SMIL via optimization to predicted 2D keypoints
- Compare this training-free 3D baseline against the learned models
This would demonstrate whether end-to-end learning actually provides value over optimization-based fitting given good 2D predictions.
4. **Limited Technical Novelty**
The contribution is primarily the dataset itself, not methodological innovation. Annotation pipeline is standard 2D COCO keypoints + 3D model fitting with human verification.

**Questions:**

1. The annotation rejection rate is quite high. What are the typical failure case reasons?
2. Only 11,642/108,902 images have 3D annotations. Does it introduce bias toward easier poses?
3. How does face blurring affect keypoint detection accuracy?

**Details Of Ethics Concerns:**

Dataset includes images of infants. Part of the dataset is obtained from YouTube videos.

---

### Official Review · Reviewer_2oKV · 2025-10-28

**Soundness:** 3
**Presentation:** 3
**Contribution:** 1
**Rating:** 2
**Confidence:** 2

**Summary:**

This paper introduces a dataset for the pose estimation of infants called InfantNet. It consists of 109K images with 2D keypoint labels, of which, 12K also have 3D ground-truth poses. This dataset is almost 10X larger than the second largest infant pose estimation dataset called MINI-RGBD. A benchmark is setup by running SOTA pose estimation methods on the dataset. The paper also introduces a pipeline to directly predict SMIL parameters from RGB inputs.

**Strengths:**

- Infant pose estimation datasets are needed, since adult pose predictors perform poorly when tested on infants (out of distribution)
- I like Table 3, which runs pre-trained models on InfantNet alongside models fine-tuned on InfantNet. The fine-tuned models are significantly more accurate, which demonstrates that existing predictors do not generalize well to infants.

**Weaknesses:**

- The ML contributions are minimal. This paper is the first to directly regress SMIL parameters from RGB (as far as I know), but this type of direct regression has been done for its adult analogue SMPL.
- The dataset itself is useful, unfortunately I don't think the wider ICLR community would find it valuable. I thus recommend a vision venue, rather than a learning venue like ICLR.

**Questions:**

None

---

### Official Review · Reviewer_mQUN · 2025-10-31

**Soundness:** 3
**Presentation:** 4
**Contribution:** 3
**Rating:** 8
**Confidence:** 4

**Summary:**

This paper introduces the InfantNet dataset, which includes images of infants (6-18 months) during typical play scenarios. Each image is annotated with 2D COCO keypoints for each joint and also a 3D-aligned SMIL model (that captures shape, pose, and position). Videos are sourced from 10 YouTube videos and 15 in-home multi-camera recordings. Ground-truth 2D keypoints are extracted automatically then verified/refined by human annotators. Ground-truth 3D SMIL parameter fittings are done automatically by a end-to-end trained neural network (briefly described; not evaluated) following human verification.

Results are given for both 2D and 3D pose estimation using several SOTA methods. In all cases, fine-tuning adult-trained models on the training subset of the InfantNet dataset leads to substantial improvements in accuracy, indicating a substantial domain gap between the two populations.

**Strengths:**

This is a strong, well-written paper introducing a unique dataset. Previous infant pose datasets were limited in size, compared to adult datasets. Size of the dataset is impressive. 100K RGB images of infants (6-18 mos) with 17-keypoint 2D annotations, and a subsetof 11.6K images that include SMIL-based 3D pose/shape annotations. The dataset appears to have been carefully prepared and includes rich, high-quality ground-truth annotations of 2D and 3D pose. Data collection and preparation are clearly described. Benchmark evaluation uses several SOTA methods for both 2D and 3D and demonstrates the value of fine-tuning on infant data. Evaluation appears to be fair, with no obvious methodological errors (e.g., separate videos are used for train/test). Collection of the HomeVisit dataset required carefully calibrated multiview video (4 cameras/views with both intrinsic and extrinsic calibration) to fit SMIL 3D model parameters to create the ground truth data. Overall, this is a valuable contribution.

**Weaknesses:**

- The end-to-end neural network model introduced to regress the SMIL parameters does not appear to be evaluated/validated in this paper (unless I missed it)

- A few minor typos to report:
  - Minor: Line 248 has a misplaced opening parenthesis in "CVAT Contributors (2018)"
  - Minor: Line 377 grammatical error in "checkpoint.... achieve"
  - Minor: capitalization of acronyms in manuscript titles within the reference list would improve readability (e.g., "rgb-d")

**Questions:**

1) The examples shown in Fig1 are convincing that the annotations align with the source image. However, the 2nd-last one (child in a doorway) highlights one potential issue: for occluded limbs, how are the annotations confirmed and can annotations of occluded limbs be considered "ground truth" (line 94)? For this specific example, the child's occluded forearm is likely reaching up or level to grasp the door frame, whereas the 3D model 'imagines' this occluded limb as continuing straight down. This leads to 2 questions: 1) How were occluded limb annotations confirmed/optimized? 2) Did you provide a measure of confidence over the annotations to indicate regions (e.g. limbs) with lower confidence?

2) Related to my first question, Line 260: Ground truth COCO pose assignments included a data quality tag of visible, occluded, not present. Do any of the prediction methods evaluated predict these data quality tags when estimating pose keypoints from an image?

3) The authors introduce a neural network pipeline for regressing SMIL (3D pose information) directly from 2D images. While the "optimization" method is briefly described, it is not evaluated in the present paper (unless I missed it). Its predictions appear to be used or rejected following qualitative assessment by human verifiers (Section 3.3). Is the proposed method evaluated in the present study?

4) Line 266: Presumably SAM2 generated both the bounding box and the segmentation mask (shown in Fig3)? No mention of saving the segmentation mask in this paragraph. Similarly, the "Final bounding box validation" paragraph does not mention verifying segmentation masks. Figure 3 shows segmentation masks, but it is not entirely clear whether those masks are included in the released dataset.

5) Line 297: Is the shape prior age-dependent, weight-dependent, or neither? Guessing no...

6) The results illustrated in Fig. 4 suggest that the model sometimes converges to an approximately correct pose estimate (where measuring MPJPE makes sense) in some cases, but completely fails to provide a reasonable pose in other cases. This suggests that evaluation may benefit from a dual-stage approach: percentage of cases where a "reasonable" prediction was rendered by the model, and then MPJPE for those cases where it converged. Reporting the mean across all cases may be heavily skewed by completely incorrect pose estimates.

7) Line 253 indicates that human annotators removed subjects older than 16 months. Should this be 18 months, since the dataset is described as containing 6-18 month infants? Also, presumably, the subject's age was estimated by the annotator? If the age was known a priori, then only eligible subjects would have been included. This could be briefly clarified in the text.

---

### Official Review · Reviewer_6bzK · 2025-11-04

**Soundness:** 3
**Presentation:** 2
**Contribution:** 4
**Rating:** 8
**Confidence:** 5

**Summary:**

This paper introduce an interesting Infant pose dataset. The dataset has 108,902 RGB images of infants aged 6-18 months.

**Strengths:**

Proposed InfantNet dataset, both contains 2D and 3D data, is a novel and large-scale real image dataset for infant pose/shape estimation, filling the gap in the relevant field.

The end-to-end SMIL-based annotation pipeline is well-designed by combining automatic models with human verification.

Experiments contains comprehensive 2D and 3D benchmarking across multiple architectures, providing useful baselines for future work, which is very thorough.

**Weaknesses:**

1. The pipeline mainly combines existing tools (RTMO-L, SAM2, SMIL) without introducing new models.

2. Data sources only comes from YouTube and HomeVisit, which lacks diverse environments and may introduce selection bias.

3. The pipeline is based on SMIL that adapting adult SMPL template. So its shape space may still not well express infant physiological structural differences, affecting the upper limit of 3D accuracy.

4. 3D labels are generated via SMIL fitting without multi-view or mocap comparison, leaving label accuracy uncertain.

5. In the benchmark part of 2D pose estimation models, many important works are not compared, e.g. AggPose (IJCAI 2022), the first infant 2D pose estimation model arround the world, only discuss in the related works but not compared; Sapiens (ECCV 2024), the SOTA human pose model.

6.  17 keypoints for infant don't have strong clinical value. This is because infants have a richer range of gross motor skills compared to adults, more keypoints from hands and feet will be helpful for downstream applications. A better design is the 21 keypoints from AggPose or more keypoints from Sapiens. It would be better if you can use Sapiens in the automatic annotation workflow.

**Questions:**

See weaknesses for more details. If the authors can successfully include more discussion on the points we suggest, we will remain our score.

**Details Of Ethics Concerns:**

Use Youtube data.

---

### Author Response · Authors · 2025-11-14

We sincerely thank all reviewers for the thoughtful and constructive feedback. The suggestions are extremely helpful for improving both the dataset and the presentation. We especially appreciate the positive assessments and detailed comments from reviewers **6bzK** and **mQUN**, who **highlighted the dataset’s scale, annotation quality, comprehensive benchmarking, and provided useful baselines for future work**.

Regarding the **ethical concerns**: all HomeVisit data were collected under IRB approval with written informed consent, and the YouTube subset was handled with strict privacy safeguards—including face masking, identity removal, and use only of derived data. In the revised version, we will add more explicit clarifications and documentation to ensure full transparency of these procedures.

Regarding **validation of the 3D annotations**: we performed two-stage quality assurance, including (1) multi-view mesh rendering with manual filtering of inconsistent or deformed registrations, and (2) reprojection-error filtering using a 16px threshold across all calibrated views. These steps help ensure the reliability of the curated 3D annotations used for model training and evaluation.

Given reviewer **2oKV**’s observation that the work may be better aligned with a vision-focused venue, we agree with this direction. We plan to revise the work and resubmit it to a computer vision conference, incorporating the reviewers’ recommendations and suggested improvements.

We again thank all reviewers for their valuable feedback. We appreciate the time and effort invested in evaluating our work and will carefully implement the suggested revisions to prepare a stronger version for a vision venue.

---

### Note · Authors · 2025-11-14

I have read and agree with the venue's withdrawal policy on behalf of myself and my co-authors.